# Protective Effects of *Hippophae rhamnoides* L. Phenylpropanoids on Doxorubicin-Induced Cardiotoxicity in Zebrafish

**DOI:** 10.3390/molecules27248858

**Published:** 2022-12-13

**Authors:** Gang Li, Ming Chu, Yingying Tong, Yuexin Liang, Shenghui Wang, Chengjun Ma, Zhenhua Wang, Wenna Zhou

**Affiliations:** 1Center for Mitochondria and Healthy Aging, College of Life Sciences, Yantai University, Yantai 264005, China; 2Department of Pharmaceutical Engineering, School of Life and Health Sciences, Huzhou College, Huzhou 313000, China

**Keywords:** sea buckthorn, zebrafish, doxorubicin, cardiotoxicity, mitochondria

## Abstract

*Hippophae rhamnoides* L. is a deciduous shrub that contains many unique bioactive substances. This sea buckthorn possesses anticancer, antioxidant, anti-inflammatory, and cardiovascular protective properties. Herein, the effects of phenylpropyl compounds extracted from *H. rhamnoides* L. on doxorubicin (Dox)-induced cardiotoxicity were evaluated in zebrafish. Cardiac injury in zebrafish was induced using 35 μM Dox for 96 h, and 30 μM phenylpropanoid compounds were used as the protective treatment. The cardioprotective effects and mechanisms of the four phenylpropanoids were investigated using microscopy, behavioral analysis, acridine orange staining, western blotting, flow cytometry, and real-time quantitative polymerase chain reaction. The extracted phenylpropanoids could significantly relieve Dox-induced cardiac injury in zebrafish and inhibit cardiomyocyte apoptosis. The mechanisms of action were mainly related to the stability of mitochondrial biogenesis and function maintained by phenylpropanoids in zebrafish. To our knowledge, this is the first report on the protective effect of sea buckthorn against myocardial injury in zebrafish. Our findings provide support for the further research and development of sea buckthorn and its components.

## 1. Introduction

Doxorubicin (Dox) is an anthracycline with broad-spectrum antitumor properties and is widely used to treat many types of cancer [1,2]. However, it has shown significant cardiotoxicity in clinical settings and can cause serious cardiac injuries [3]. Studies in patients with pediatric acute lymphoblastic leukemia show that Dox-induced cardiac injury can occur during and several years after the completion of drug therapy, progressing from initial asymptomatic left ventricular dysfunction to dilated cardiomyopathy, congestive cardiac failure, and even death [4,5]. The cardiac injury accompanying the therapy is positively correlated with the cumulative dose of Dox, thus greatly limiting its use in clinical practice. Moreover, Dox can induce cardiomyocyte apoptosis through various mechanisms, the more prominent of which is the mitochondrial pathway. Mitochondrial dysfunction could serve as an important marker to detect Dox-induced cardiotoxicity [6]. The energy supply of the heart is mainly derived from mitochondria (90%), and the heart tissue is rich in mitochondria. Dox can accumulate specifically in the mitochondria and eventually lead to cardiomyocyte apoptosis [7].

The sea buckthorn *Hippophae rhamnoides* L. is a deciduous shrub belonging to the family *Elaeagnaceae*. Its extract contains many unique bioactive substances with high medicinal value and development potential. Pharmacological studies of sea buckthorn have demonstrated its anticancer, antioxidant, antiapoptotic, antibacterial, anti-inflammatory, wound-healing, and cardiovascular-protective properties [8,9,10,11,12]. Among them, its cardioprotective effects are particularly significant. Sea buckthorn pulp oil can ameliorate myocardial ischemia-reperfusion injury by modulating the expression of Akt-eNOS and IKKβ/NF-κB [13]. Two alkaloids from sea buckthorn were shown to inhibit Dox-induced injury in H9C2 cardiomyocytes by improving mitochondrial function [14]. The cardioprotective effects of sea buckthorn and some of its extracts have been demonstrated. However, to our knowledge, studies investigating the myocardial-protective effects of sea buckthorn and its extracts using an in vivo model of zebrafish have not been reported. Zebrafish has 87% genetic similarity with humans, far exceeding that of commonly used experimental rodent models, thereby serving as a novel in vivo model owing to its unique advantages [15].

The four phenylpropanoid monomers p-coumaric acid (P1), chlorogenic acid (P2), caffeic acid (P3), and ferulic acid (P4) (Figure 1) were isolated and purified from *H. rhamnoides* L. seeds following the method reported in our previous study. In this study, the cardioprotective effects of these four phenylpropanoids were assayed and their mechanisms of action were investigated using a Dox-induced cardiac injury model of zebrafish. Exploring new and effective cardioprotective agents from sea buckthorn and using these compounds to prevent Dox-induced cardiac injury has important implications for the prognosis of patients with tumors.

## 2. Results

### 2.1. Effects of H. rhamnoides L. Phenylpropanoids on Dox-Induced Cardiac Injury in Zebrafish

Zebrafish larvae treated with Dox alone showed significant abnormalities in cardiac morphology and exhibited a low heart rate. These cardiac injuries were effectively improved by pretreatment with *H. rhamnoides* L. phenylpropanoids. Figure 2A shows the representative photomicrographs. Zebrafish larvae in the Dox group showed pericardial edema, hemorrhage, and linearization of heart morphology compared with those in the control group. All larvae in the phenylpropanoid-treated groups showed an improvement in cardiac morphological abnormalities compared with those in the Dox group. Pericardial areas were quantified using ImageJ and histograms (Figure 2B) were obtained. Larvae in the Dox group had a significantly larger pericardial area than those in the control group (*p* < 0.01), and the pericardial area decreased significantly in larvae in the four phenylpropanoid-treated groups compared with those in the Dox group (*p* < 0.01). The distance between the sinus venosus (SV) and bulbus arteriosus (BA) reflects the degree of drug effect on the cardiac-cyclization ability of zebrafish. As shown in Figure 2C, the SV-BA distance of zebrafish larvae in the Dox group was significantly greater than that of larvae in the control group (*p* < 0.01), and the heart showed abnormal linearization. The SV-BA distance was significantly reduced in all phenylpropanoid-treated groups (*p* < 0.01) and the degree of cardiac cyclization showed improvement. In addition, as shown in Figure 2D, the heart rate of larvae in the Dox group reduced significantly (*p* < 0.01). Treatment with phenylpropanoids led to a significant increase in the heart rate of larvae compared with those treated with Dox (*p* < 0.05, *p* < 0.01).

### 2.2. Effects of H. rhamnoides L. Phenylpropanoids on Locomotor Activity in Zebrafish

To further investigate the effect of Dox on zebrafish larvae, their locomotor activity was examined. The representative locomotor trajectories of each group are shown in Figure 3A. Zebrafish larvae in the control group were more active and had a good locomotor ability, whereas those in the Dox group showed a significant decrease in locomotor ability compared with the control larvae. The four phenylpropanoid-treated groups exhibited different degrees of improvement in the Dox-induced decrease in zebrafish motility. The distance moved by each zebrafish was quantified (Figure 3B). The distance moved by zebrafish in the Dox group was significantly lesser (*p* < 0.01), whereas the distance moved by larvae in the phenylpropanoid-treated groups was significantly higher compared with those in the Dox group. P1 + Dox and P4 + Dox treatment, especially, led to significant differences in movement. Therefore, Dox-induced cardiac damage affected zebrafish locomotion, and pretreatment with phenylpropanoids exerted a protective effect and prevented the decrease in locomotion.

### 2.3. Effects of H. rhamnoides L. Phenylpropanoids on Dox-Induced Apoptosis in Zebrafish

There was a significant improvement in Dox-induced cardiac injury after pretreatment with *H. rhamnoides* L. phenylpropanoids. To investigate the mechanism of action in improving cardiac injury, acridine orange (AO) staining was performed and the expression of the apoptosis-related protein cleaved caspase-3 was analyzed in each group. Figure 4A shows the representative AO staining images. Compared with the larvae in the control group, those in the Dox group showed more granular yellow-green fluorescence in the heart. The fluorescence was reduced in zebrafish larvae in the phenylpropanoid-treated groups, especially in the P1 + Dox and P4 + Dox groups, compared with those in the Dox group. Cleaved caspase-3 expression in zebrafish larvae was determined using western blotting. When apoptosis occurs, caspase-3 is activated to become cleaved caspase-3. As shown in Figure 4C, cleaved caspase-3 expression increased in the Dox group compared with that in the control group. Cleaved caspase-3 expression in the phenylpropanoid-treated groups decreased in all groups except the P2 + Dox group, and its expression was significantly lower in the P1 + Dox and P4 + Dox groups (*p* < 0.05). The results indicated that Dox induced cardiomyocyte apoptosis, whereas phenylpropanoid treatment inhibited apoptosis to some extent by downregulating cleaved caspase-3 expression.

### 2.4. Effects of H. rhamnoides L. Phenylpropanoids on Mitochondrial-Related Assays in Zebrafish

The effects of drugs on mitochondrial quality and function were investigated by determining the mitochondrial copy number, mitochondrial membrane potential (MMP), and ATP levels. The mitochondrial DNA copy number of zebrafish was determined using real-time quantitative PCR (rT-qPCR) and the results are shown in Figure 5A. The mitochondrial copy number was significantly decreased in the Dox group versus the control group (*p* < 0.01). The phenylpropanoid-treated groups exhibited improvements in mitochondrial copy number, with significant increases in the P1 + Dox and P4 + Dox groups compared with that in the Dox group (*p* < 0.01). In the MMP assay, carbonyl cyanide 3-chlorophenylhydrazone (CCCP) is used as a positive control to induce a decrease in MMP. Changes in MMP in zebrafish were determined using the fluorescent probe JC-1, and the results are shown in Figure 5B. MMP in both Dox and CCCP groups decreased significantly compared with that in the control group (both *p* < 0.01). The MMP increased in both P1 + Dox and P4 + Dox groups compared with that in the Dox group, with the increase in the P4 + Dox group being significantly different (*p* < 0.05). As shown in Figure 5C, ATP levels decreased significantly in the Dox group compared with those in the control group (*p* < 0.01). Among the phenylpropanoid-treated groups, the improvement was more significant in the P1 + Dox and P4 + Dox groups, and a significant increase in ATP level was observed in these groups compared with that in the Dox group (*p* < 0.01). The results suggested that the protective effect of *H. rhamnoides* L. phenylpropanoids against Dox-induced cardiotoxicity was mitochondria-related.

### 2.5. Effects of H. rhamnoides L. Phenylpropanoids on Mitochondrial Dynamics–Related Proteins

Mitochondrial dynamics is the basic mitochondrial quality control method that involves mitochondrial division and fusion. Mitochondria maintain normal morphology and number through division and fusion to ensure proper physiological functions. Figure 6A,B show the expression and analysis of the mitochondrial division protein Drp1 and the mitochondrial fusion protein Mfn1, respectively, in each group. Overall, the expression of Drp1 increased slightly and that of Mfn1 decreased in the Dox group compared with the corresponding expression in the control group. This change may affect the dynamic balance of mitochondrial division and fusion. Drp1 expression in the P1 + Dox and P4 + Dox groups decreased, whereas Mfn1 expression in the P3 + Dox and P4 + Dox groups increased; however, these changes were not statistically significant. These findings suggested that the phenylpropanoids could contribute to some extent in maintaining the dynamic homeostasis in mitochondria, but this effect was not sufficiently stable.

### 2.6. Effects of H. rhamnoides L. Phenylpropanoids on Mitochondrial Biogenesis in Zebrafish

Mitochondrial biogenesis is the process of forming new mitochondria within cells and is a pathway for the quality-control mechanism for mitochondria. The central link of its regulation mechanism is PGC-1α and its downstream regulatory factors such as NRF1 and Tfam. Figure 7A shows the protein bands of PGC-1α, NRF1, and Tfam obtained using western blotting. Figure 7B–D show the analysis of PGC-1α, NRF1, and Tfam expression in each group, respectively. PGC-1α, NRF1, and Tfam expression in the Dox group decreased compared with that in the control group; among them, the expression of NRF1 and Tfam was significantly different (*p* < 0.05), whereas that of PGC-1α was highly significantly different (*p* < 0.01). These results indicated that Dox inhibited mitochondrial biogenesis. All groups showed an elevated trend after phenylpropanoid treatment. Among them, PGC-1α expression increased significantly in the P4 + Dox group (*p* < 0.05), whereas NRF1 expression increased significantly in the P1 + Dox and P4 + Dox groups (*p* < 0.05). Tfam expression increased highly significantly in all four protected groups (*p* < 0.01). Thus, phenylpropanoids could inhibit Dox-induced cardiotoxicity by activating mitochondrial biogenesis and maintaining mitochondrial quality.

## 3. Discussion

Dox is a broad-spectrum antibiotic widely used to treat various cancers; however, it can cause severe dose-dependent cardiac damage in clinical settings [16]. Therefore, identifying measures to inhibit cardiotoxicity is crucial. Our research group has conducted several studies on the active components of sea buckthorn, including flavonoids and alkaloids [17,18]. In the current study, the cardioprotective effects of four phenylpropanoid monomers extracted from *H. rhamnoides* L. seeds were evaluated using a Dox-induced cardiac injury model of zebrafish. Numerous studies have shown that sea buckthorn contains bioactive substances that exert ameliorative effects in patients with cardiovascular diseases [19]. Therefore, we evaluated whether the four phenylpropanoid monomers extracted from *H. rhamnoides* L. could exert cardioprotective agents and prevent Dox-induced cardiotoxicity.

In this study, Dox was used for modeling. The zebrafish larvae in the Dox group showed significant swelling and hemorrhage of the pericardium, enlargement of the yolk sac, linearization of cardiac malformations, increased SV-BA spacing, and reduced heart rate compared with those in the control group, among other signs of cardiac injury. Pretreatment with phenylpropanoids significantly improved Dox-induced cardiac injury in zebrafish. To further investigate the effect of Dox on zebrafish larvae, their locomotor activity was examined. The results showed that the motility of zebrafish in the Dox group was reduced and that phenylpropanoid treatment exerted a protective effect and prevented the reduction in motility. Related behavioral studies have been reported less frequently, but our finding suggested that Dox-induced cardiac injury affects the locomotor ability to some extent. Apoptosis of zebrafish cardiomyocytes was determined using AO dye liquor. Yellow-green fluorescent spots in the heart increased significantly in larvae in the Dox group compared with those in the normal control group. The yellow-green spots represent the production of apoptotic vesicles [20]. All four phenylpropanoids reduced the production of yellow-green fluorescent spots in this experiment, with P1 and P4 being more effective. Caspase-3 is an important protease that participates in apoptosis. The expression of cleaved caspase-3 protein was determined, and the activation of caspase-3 was found to be significantly increased in larvae in the Dox group. Cleaved caspase-3 expression was found to be decreased in the phenylpropanoid-treated groups and caspase-3 activation was effectively inhibited. Among them, P1 and P4 had better effects, which were consistent with the results from AO staining. It has been reported that the *Ganoderma atrum* polysaccharide, PSG-1, inhibits Dox-induced apoptosis in mice by increasing the expression of the anti-apoptotic protein and decreasing that of the pro-apoptotic protein, thereby suppressing Dox-induced cardiotoxicity [21]. Apoptosis is positively correlated with Dox-induced cardiotoxicity. Cardiomyocytes contain abundant mitochondria, which play an important role in maintaining normal cell homeostasis. Dox-induced mitochondrial damage is the central mechanism of the cardiac action of Dox [22]. Therefore, further mitochondria-related assays were performed. A significant decrease in mitochondrial copy number, MMP, and ATP levels was observed in the Dox group, indicating that Dox could affect mitochondria and impair mitochondrial function. In contrast, the four phenylpropanoids ameliorated the Dox-induced abnormal findings, as depicted by the mitochondria-related assays. These results indicated that phenylpropanoid treatment could ensure the normal functioning of mitochondria to some extent. In terms of mitochondrial dynamics, Dox decreased Mfn1 expression and increased Drp1 expression to some extent. Improving this abnormality in protein expression could rebalance the mitochondrial dynamics and reduce Dox-induced cardiotoxicity [23,24]. Our findings indicated that the phenylpropanoid-treated groups could maintain the balance of mitochondrial dynamics to some extent, but the effect was not significant. Detection of mitochondrial biogenesis–related indicators revealed that treatment with Dox led to a decrease in PGC-1α and that its downstream NRF1 and Tfam were also decreased. This would ultimately result in a decrease in mitochondrial copy number, consistent with previously published results. The expression of related proteins increased to different degrees after treatment with phenylpropanoids. Among them, treatment with P4 led to a significant difference in the improvement in the protein expression of PGC-1α, NRF1, and Tfam, whereas treatment with P1 led to significant differences in the improvement of downstream NRF1 and Tfam expression profiles. Although treatment with all four phenylpropanoids led to some extent of improvement in the expression of mitochondrial biogenesis-related proteins, the effects of P1 and P4 were particularly pronounced. The CO/HO system ameliorated Dox-induced cardiotoxicity by reversing Dox-induced inhibition of mitochondrial biogenesis [25]. Cannabidiol protects against Dox-induced cardiotoxicity by maintaining stable mitochondrial function and regulating mitochondrial biogenesis [26]. Endothelin-converting enzyme-1 ablation attenuated Dox-induced cardiomyopathy by preventing impaired mitochondrial biogenesis [27]. Related studies have shown that the activation of mitochondrial biogenesis is effective in protecting against Dox-induced cardiotoxicity. The four phenylpropanoids evaluated in our study inhibited Dox-induced cardiotoxicity mainly by activating mitochondrial biogenesis and maintaining mitochondrial function.

## 4. Materials and Methods

### 4.1. Chemicals

Dox (CAS: 25316-40-9; Aladdin; Purity: 98%) was prepared as a 5 mM stock solution in ultrapure water and stored at −20 °C until use. The phenylpropanoid monomers p-coumaric acid (P1), chlorogenic acid (P2), caffeic acid (P3), and ferulic acid (P4) were isolated and purified from *H. rhamnoides* L. seeds obtained from the Research and Development Group of Ecological and Economic Plant Resources of Qinghai-Tibet Plateau, Northwest Plateau Institute of Biology, Chinese Academy of Sciences. Their purity was determined to be >95% using high-performance liquid chromatography. The compounds were detected by NMR (1H, 13C) with the nuclear magnetic resonance spectrometer (JNMECZ400S, JEOL, Akishima, Japan) and the relevant information can be found in the Appendix A. Stock solutions of P1, P2, P3, and P4 were prepared in dimethylsulfoxide (100 mM) and stored at −20 °C. The stock solutions were diluted with E3 medium (5 mM NaCl, 0.17 mM KCl, 0.4 mM CaCl_2_, and 0.16 mM MgSO_4_) prior to use.

### 4.2. Zebrafish Maintenance

Zebrafish (*Danio rerio*, AB line), gifted by Professor Bo Zhang of Peking University, were maintained by the Center for Mitochondria and Healthy Aging, Yantai University. Adult zebrafish were reared and maintained in a recirculating aquaculture system (Aisheng, Beijing, China), and the water in the system was maintained separately at 28.0 ± 0.5 °C, pH 7.0–8.0, conductivity 450–550 μs. Zebrafish were fed freshly hatched artemia twice daily and exposed to a 14-h/10-h light/dark cycle to provide optimal conditions for normal growth and development. Mature male and female zebrafish were transferred into a mating box that was separated by a partition. The partition was removed 30 min after the 14-h photoperiod began the next day. After 30 min, the embryos were collected from the mating box and transferred to Petri dishes containing E3 medium for incubation under the light/dark cycle.

### 4.3. Zebrafish Treatment

Healthy zebrafish embryos were selected under a stereomicroscope at 6 h post fertilization (hpf) and transferred to 6-well culture plates. Embryos were randomly assigned to wells (30 per well) and incubated with 3 mL of the test solutions. The normal control group was incubated with E3 medium. The model group was incubated with 35 μM Dox solution. The phenylpropanoid-treated groups were first pretreated with 30 μM of the respective phenylpropanoids for 1 h and then co-treated with the phenylpropanoid and Dox. Three replicate wells were used for each group. Dead embryos were removed and the solution in wells was replaced every 24 h. The experiment was terminated after 96 h.

### 4.4. Observation of Pericardial Morphology

Experimental endpoints were reached after 96 h of drug treatment. Zebrafish larvae were washed three times with PBS. The treated zebrafish larvae were anesthetized using 0.016% tricaine [28]. Eight zebrafish larvae from each group were randomly selected and observed and imaged using microscopy to determine the effects of drug treatment. To facilitate inspection, 0.4% methylcellulose solution was added dropwise to fix the posture of the zebrafish larvae. Images of the lateral views were collected using a DMi1 optical inverted microscope (Leica, Wetzlar, Germany) [29].

### 4.5. Quantification of the Pericardial Area

Images of zebrafish larvae were collected as described in Section 4.4. Pericardial images of zebrafish were captured using a DMi1 optical inverted microscope and the shape of the pericardial edema was outlined. Lastly, the area of the outlined figure was calculated [30].

### 4.6. Quantification of Cardiac Cyclization

Zebrafish hearts start to develop at 6 hpf and the development is complete at 48 hpf. The degree of cardiac looping is a critical step in the development of the heart. SV-BA distance can be used as a key index to quantify the degree of cardiac cyclization. The images of zebrafish larvae were collected as described in Section 4.4. Based on the length between the two points from SV to BA in the figure, the actual distance in proportion was calculated [31].

### 4.7. Heart Rate Measurement

Heart rate is a direct reflection of the extent that the drug affects the heart. At the experimental endpoint, zebrafish larvae were washed three times with PBS and anesthetized using 0.016% tricaine solution. Eight anesthetized zebrafish larvae from each group were randomly selected and placed on slides, and the number of heartbeats was observed and recorded for 15 s using a microscope [32].

### 4.8. Locomotor Activity Assessment

The treated zebrafish larvae were transferred to 96-well plates (1 zebrafish larva per well) and 200 μL E3 medium was added. Zebrafish movements were recorded using a video camera (Logitech, Shanghai, China) and the distance moved by each zebrafish was quantified using SMART 3.0 software (Panlab Harvard Apparatus, Horriston, MA, USA). In this experiment, 30 larvae were tested in each group at a time, and the locomotor activity of each group of larvae was assessed using the 10 min movement distance and 60 s movement trajectory graph of zebrafish larvae.

### 4.9. AO Staining

AO (Sigma-Aldrich, Shanghai, China) is used to stain apoptotic cells. Chromatin in apoptotic cells solidifies or breaks into pieces of varying sizes to form apoptotic bodies, which are stained by AO to exhibit a dense, granular, yellow-green fluorescence. AO dye was prepared to a concentration of 5 mg/mL and diluted to 5 µg/mL before use [33]. After drug treatment, the zebrafish larvae were washed 3 times with PBS, transferred to the diluted AO dye solution, and incubated for 30 min at room temperature while being protected from light. After staining, the dye was aspirated and the larvae were rewashed 3 times with PBS. Eight zebrafish larvae were randomly selected from each group and photographed using a DMi8 fluorescence microscope (Leica, Wetzlar, Germany) as described in Section 4.4.

### 4.10. Western Blotting

At the experimental endpoint, zebrafish larvae were washed 3 times with PBS and transferred to Eppendorf (EP) tubes on ice. Sixty zebrafish larvae were selected from each group, and 100 μL RIPA lysis buffer (Beyotime, Shanghai, China) containing 1% phenylmethylsulfonyl fluoride (Beyotime, Shanghai, China) was added after the aspiration of PBS. Mechanical homogenization was performed on ice and centrifuged (14,000× *g* at 4 °C for 15 min) after lysis for 30 min at 4 °C [34]. The supernatant was collected for subsequent experiments. Total protein concentration was determined using a bicinchoninic acid protein-detection kit (Beyotime, Shanghai, China) and complete protein quantification. Protein bands of the prepared samples were obtained using sodium dodecyl sulfate-polyacrylamide gel electrophoresis, which were then transferred to polyvinylidene fluoride membranes. The membranes were blocked with 5% skim milk for 1 h, washed 3 times with TBST, and incubated overnight at 4 °C with the following primary antibodies: rabbit anti-caspase-3 (1:1000, CST, Boston, MA, USA), rabbit anti-Drp1 (1:1000, CST, MA, USA), rabbit anti-Mfn1 (1:1000, CST, MA, USA), rabbit anti-Tfam (1:1000, Abcam, Cambridge, MA, USA), rabbit anti-PGC-1α (1:1000, CST, MA, USA), and rabbit anti-tubulin (1:1000, CST, MA, USA). The membranes were washed 3 times with TBST and subsequently incubated with the horseradish peroxidase–coupled secondary antibody (1:2000, NVSI, Shanghai, China) for 1 h. After 3 washes with TBST again, the protein bands were detected using enhanced chemiluminescence reagent (Beyotime, Shanghai, China). The relative gray-scale values were analyzed using ImageJ software [35].

### 4.11. Assay to Determine ATP Content

Sixty zebrafish larvae that were treated for 96 h were selected from each group and placed in EP tubes. After aspirating the excess liquid in the tubes, 150 μL of lysis solution was added to each tube. An electric homogenizer was used for homogenization and the samples were centrifuged (12,000× *g* at 4 °C for 5 min) after completion of lysis; the supernatant was retained. ATP content in zebrafish tissues was measured using an ATP assay kit (Beyotime, Shanghai, China).

### 4.12. Assay to Determine MMP

Sixty zebrafish larvae that were treated for 96 h were selected from each group and placed in EP tubes. After aspirating the excess liquid, 400 μL of 0.2% collagenase II (Biodee, Beijing, China) and 100 μL of 0.05% trypsin (Biodee, Beijing, China) were added to each tube. The tubes were placed in a water bath at 37 °C for 60 min and 500 μL of 10% fetal bovine serum (Tianhang, Zhejiang, China) was added to terminate the digestion. Cell suspensions obtained after filtration were mixed with 0.4% trypan blue (Solarbio, Beijing, China) solution in a 1:9 ratio. The mixed solution was added to a hemocytometer within 3 min and cell necrosis was observed using microscopy. A live cell count of 90% or higher was required before proceeding to the next step. The qualified mixtures were centrifuged at 1000 rpm at 4 °C for 5 min and the supernatant was discarded. Precooled PBS was added and mixed well. The supernatant was discarded after centrifugation, and 10 μM of JC-1 (1:1000, Sigma-Aldrich, Shanghai, China) solution was added [36]. The samples were allowed to stand for 20 min in the dark at room temperature. The supernatant was discarded after re-centrifugation and PBS was added. Changes in MMP were determined using flow cytometry (Becton Dickinson, CA, USA).

### 4.13. Mitochondrial Copy Number Assay

Treated zebrafish embryos (60 per tube) were added to 600 μL of precooled PBS and homogenized on ice for approximately 20 s. Samples were centrifuged at 12,000 rpm at 20 °C for 1 min. Next, the supernatant was discarded and zebrafish embryonic DNA was extracted using a TIANamp Genomic DNA kit (Tiangen, Beijing, China). The concentration of the extracted DNA was determined using a microplate reader (Bioteke, Beijing, China). The DNA template concentration was adjusted to 15 ng/μL using TE buffer. The reaction solution was prepared using SYBR Premix Ex Taq TM II (TaKara Bio Inc., Beijing, China) and RT-qPCR was performed by a 7500 Fast Real-Time PCR System (Applied Biosystems, Waltham, CA, USA) [37]. The primer sequences used in the reactions are shown in Table 1. The cycling conditions were as follows: 95 °C for 10 min and 40 cycles of 95 °C for 15 s, 60 °C for 45 s, and 72 °C for 45 s. β-Actin was used as an internal reference gene and calculated using the ΔΔCt approach.

### 4.14. Statistical Analysis

Three replicates were performed for each experiment. Data are expressed as the mean ± standard deviation. Differences between groups were evaluated using one-way analysis of variance or *t*-test using GraphPad Prism 8.0.1. *p* < 0.05 was considered statistically significant.

## 5. Conclusions

The cardioprotective effects of the four phenylpropanoid monomers extracted from *H. rhamnoides* L. were evaluated using a Dox-induced cardiac injury model of zebrafish. All four phenylpropanoid monomers could alleviate Dox-induced cardiac injury in zebrafish, especially p-coumaric acid and ferulic acid. Further studies showed that the mechanism was mainly related to activating mitochondrial biogenesis and maintaining the stability of mitochondrial function. However, the involved mechanism of action is still not well understood. Additional mechanistic studies are therefore warranted to provide better support for the development and use of sea buckthorn and its components.

## Figures and Tables

**Figure 1 molecules-27-08858-f001:**
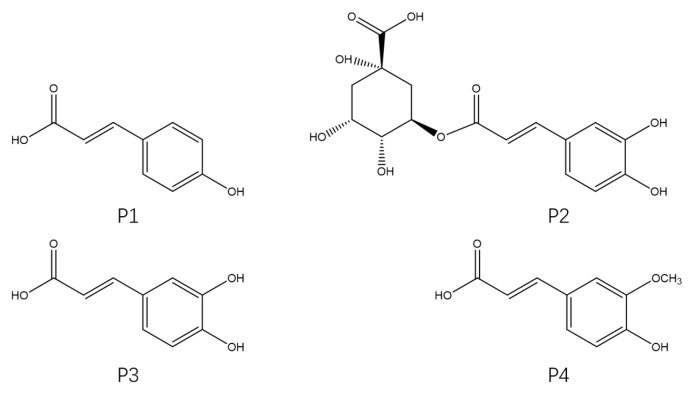
Chemical structures of the four phenylpropanoids from *Hippophae rhamnoides* L.

**Figure 2 molecules-27-08858-f002:**
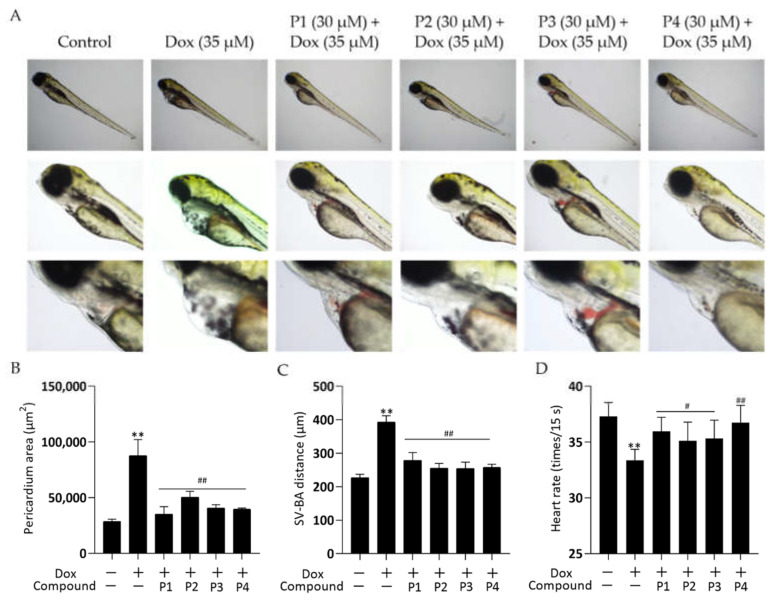
Effects of phenylpropanoids of *Hippophae rhamnoides* L. on doxorubicin-induced cardiac injury in zebrafish. (**A**) Representative photographs of zebrafish pericardium collected using a DMi1 optical inverted microscope. (**B**) Column chart of pericardial edema in each group obtained by quantifying the pericardial area using ImageJ software. (**C**) Column chart of the distance between sinus venosus and bulbus arteriosus obtained using microscopy. (**D**) Column chart of heart rate per 15 s in zebrafish larvae. ** *p* < 0.01 vs. the control group; ^#^
*p* < 0.05, ^##^
*p* < 0.01 vs. the Dox group.

**Figure 3 molecules-27-08858-f003:**
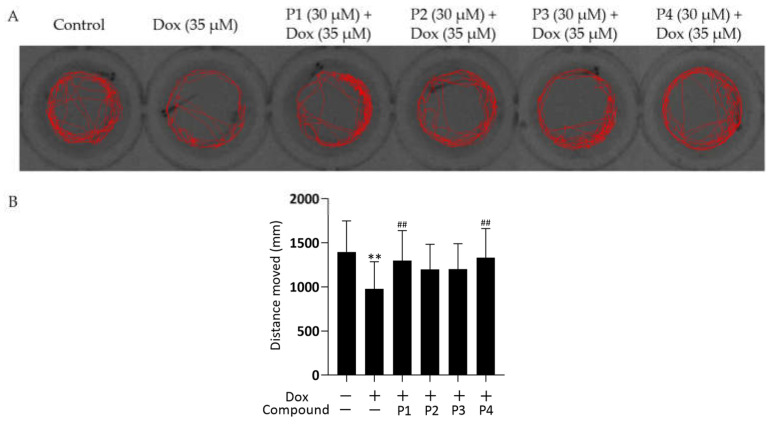
Effects of *Hippophae rhamnoides* L. phenylpropanoids on the locomotor activity of zebrafish. (**A**) Representative images of the 60 s movement trajectory of zebrafish. (**B**) Quantification of the 10 min distance moved by zebrafish. ** *p* < 0.01 vs. the control group; ^##^
*p* < 0.01 vs. the Dox group.

**Figure 4 molecules-27-08858-f004:**
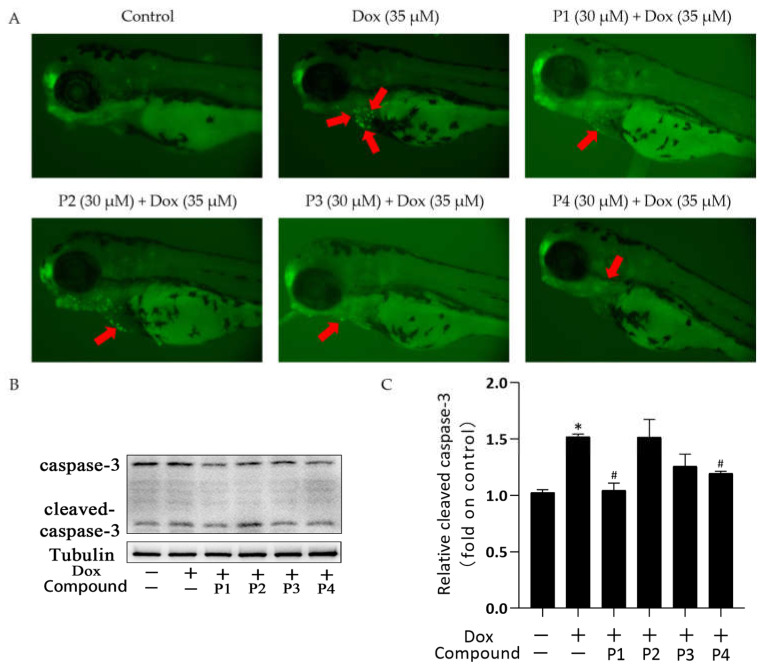
Effects of *Hippophae rhamnoides* L. phenylpropanoids on Dox-induced apoptosis in zebrafish. (**A**) Apoptosis of zebrafish cardiomyocytes observed using acridine orange staining (Arrows indicate apoptotic staining). (**B**) Cleaved caspase-3 protein expression using western blotting. (**C**) Column chart of the quantitative analysis of cleaved caspase-3 protein expression in zebrafish. * *p* < 0.05 vs. the control group; ^#^
*p* < 0.05 vs. the Dox group.

**Figure 5 molecules-27-08858-f005:**
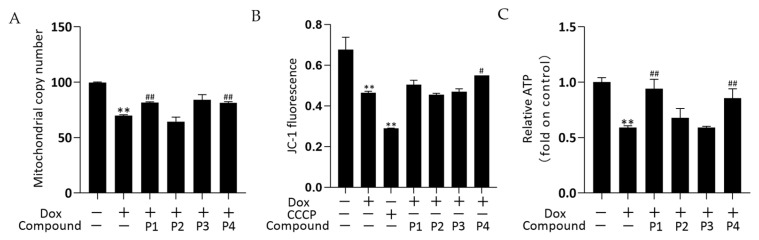
Effects of *Hippophae rhamnoides* L. phenylpropanoids on the mitochondrial function of zebrafish. (**A**) Mitochondrial copy number in each group of zebrafish obtained using real-time quantitative PCR. (**B**) Mitochondrial membrane potential of each group in zebrafish determined using flow cytometry. (**C**) ATP levels in each group of zebrafish obtained using an ATP assay kit. ** *p* < 0.01 vs. the control group; ^#^
*p* < 0.05, ^##^
*p* < 0.01 vs. the Dox group.

**Figure 6 molecules-27-08858-f006:**
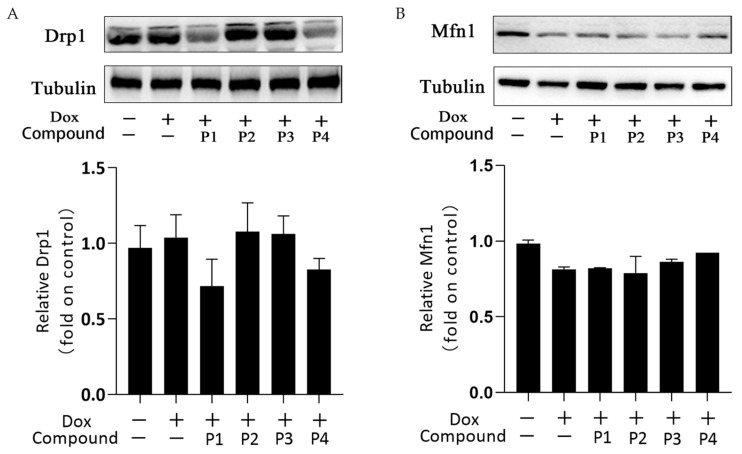
Effects of *H. rhamnoides* L. phenylpropanoids on mitochondrial dynamics–related proteins. (**A**) Expression and analysis of the mitochondrial division protein Drp1. (**B**) Expression and analysis of the mitochondrial fusion protein Mfn1.

**Figure 7 molecules-27-08858-f007:**
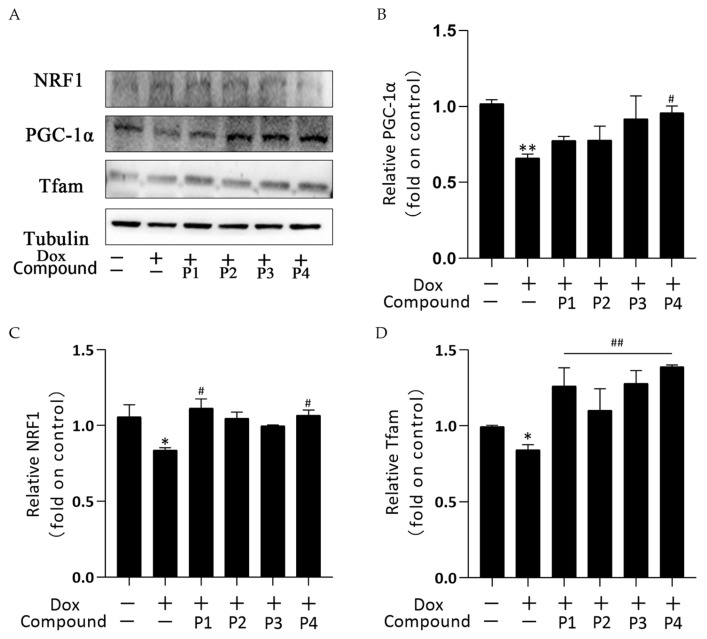
Effects of *Hippophae rhamnoides* L. phenylpropanoids on mitochondrial biogenesis in zebrafish. (**A**) PGC-1α, NRF1, and Tfam protein expression determined using western blotting. (**B**) Column chart of quantitative analysis of PGC-1α protein expression in zebrafish. (**C**) Column chart of quantitative analysis of NRF1 protein expression in zebrafish. (**D**) Column chart of quantitative analysis of Tfam protein expression in zebrafish. * *p* < 0.05, ** *p* < 0.01 vs. the control group; ^#^
*p* < 0.05, ^##^
*p* < 0.01 vs. the Dox group.

**Table 1 molecules-27-08858-t001:** Primers used for real-time quantitative PCR.

Target Gene	Primer Sequence
Mitochondrial	Forward: 5′-CAAACACAAGCCTCGCCTGTTTAC-3′
	Reverse: 5′-CACTGACTTGATGGGGGAGACAGT-3′
β-Actin	Forward: 5′-ATGGATGAGGAAATCGCTGCC-3′
	Reverse: 5′-CTCCCTGATGTCTGGGTCGTC-3′

## Data Availability

Not applicable.

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
