# Peer review of "Protective Effects of Hippophae rhamnoides L. Phenylpropanoids on Doxorubicin-Induced Cardiotoxicity in Zebrafish"

_molecules, 2022, doi:10.3390/molecules27248858_

Round 1

Reviewer 1 Report

This paper is very nicely written. A clear read. The objective and choice of experiments is clearly presented. One edit must be the removal of And from the beginning of sentences please. This occurs regularly. 

I have a deep concern about the statement that embryos were dying during this experiment and no record of the level of this death is given. This over shadows the results because I am concerned high numbers maybe dying in particular experimental units. 30 embryos were incubated but how many were used in each experiment? 30 for each I presume? This is not clear to me. Perhaps you used the same fish in multiple analyses? 

I am surprised that the discussion does not dissect out the clear differences between P1-P4. This is important and is only mentioned in the conclusion. Clearly different effects are happening and with each extract. These are very interesting result and needs to be considered more. A table may represent these differences and a score of the extracts potential, as a collection figure? 

Can you please detail any blinding and randomisation that was used in the data collection. This is important according to the ARRIVE guidelines. Please take a look at these guidelines and consider if your experiment is reproducible. For instance were the embryos in an incubator with a light cycle? 

Reviewer 2 Report

The manuscript sounds good with one exception: the substances studied were not characterized properly.

Lines 63-65: “The four phenylpropanoid monomers P-Coumaric acid (P1), Chlorogenic acid (P2), Caffeic acid (P3) and Ferulic acid (P4) (Figure 1) were isolated and purified from Hippophae rhamnoides L. seeds in our previous study.” – reference required, the structural identity of the compounds could not be verified. Moreover, in Materials and Methods, the Chemicals explored are characterized as follows:
Lines 284-289 - “The phenylpropanoid monomers P-Coumaric acid (P1), Chlorogenic acid (P2), Caffeic acid (P3) and Ferulic acid (P4) were isolated and purified from Hippophae rhamnoides L. seeds by the Research and Development Group of Ecological and Economic Plant Resources of Qinghai-Tibet Plateau, Northwest Plateau Institute of Biology, Chinese Academy of Sciences, and their purity was above 95% by High Performance Liquid Chromatography (HPLC).”  – That is not sufficient for unambiguous identification of the substances. Either chemicals should be purchased from an authoritative supplier, or authors should provide NMR (1H,  13C, HSQC, HMBC) and MS data (exact mass, preferably in both positive and negative ionisation mode) of each compound. The reference to a previously published article with that characterisation (see above) would be Ok. Alternatively, the compounds purified from the natural source could be used in all biological experiments, but the identity with reference compound (purchased from supplier) should be demonstrated with 1H NMR (two identical NMR spectra).

Typos in line: 16 (cardioCtoxicity)

Round 2

Reviewer 2 Report

The authors provided high-quality NMR data and completely characterized the structures under investigation.

There are two typo errors in NMR data description. The first is in the structure on “figure S6. Structure of chlorogenic acid (P2)”. Methyles are drawn on the structure, they should be hydroxy groups at positions 2', 3', 5' (as obvious from NMR data and figure 1 from the main text). The second typo error is in the description of methoxy H10 under “figure S12. Structure of ferulic acid (P4)”. Author wrote "3.77 (1H, s, H-10)", there should be three protons and "3.77 (3H, s, H-10)". 

The NMR spectra are unambiguous and provide final proof of the correct structures. I’m sure that the authors will correct the errors above in the final version of the manuscript.

I have no additional comments.